# Climate change, biodiversity loss, and Indigenous Peoples' health and wellbeing: A systematic umbrella review

Laura Jane Brubacher[1]*, Laura Peach[1], Tara Tai-Wen Chen[2], Sheri Longboat[3,4], Warren Dodd[1], Susan J. Elliott[2], Kaitlyn Patterson[1], Hannah Neufeld[1]

1 School of Public Health Sciences, University of Waterloo, Waterloo, Ontario, Canada, 2 Department of Geography and Environmental Management, University of Waterloo, Waterloo, Ontario, Canada, 3 School of Environmental Design and Rural Development, University of Guelph, Guelph, Ontario, Canada, 4 Department of Geography and Environmental Studies, Wilfrid Laurier University, Waterloo, Ontario, Canada

* ljbrubacher@uwaterloo.ca

**Data Availability Statement:** Data is available in the Supporting information files.

## Abstract

Indigenous Peoples worldwide are experiencing a cascade of impacts on their health and wellbeing as a result of climate change and biodiversity loss. Existing literature at the interface of climate change, biodiversity loss, and Indigenous health tells us that Indigenous Peoples are among those most disproportionately and acutely affected by these impacts. Yet, a gap exists with respect to comprehensively and critically synthesizing the impacts reported across this literature and identifying Indigenous-led responses. Guided by an Indigenous advisory group, we employed a systematic umbrella review methodology, following PRISMA guidelines, to characterize the global secondary literature (PROSPERO registration #: CRD42023417060). In so doing, we identified the proximal, intermediate, distal, and gendered impacts of climate change and biodiversity loss on Indigenous health and wellbeing as well as Indigenous-led responses. Five databases were searched for published reviews, along with a grey literature search that focused on underrepresented geographic regions in the academic literature. Two independent reviewers conducted two-stage screening, data extraction, and quality assessment of retrieved records. Basic descriptive statistics were calculated. Qualitative data were analyzed thematically, using a constant comparative approach. A total of 38 review articles met the eligibility criteria and 37 grey literature records were retrieved and included in the review. Reviews were published between 2010–2023 and geographically clustered in the Circumpolar North. Intersecting proximal, intermediate, and distal impacts were characterized as place-based and specific, and linked to colonialism as an antecedent to and driver of these impacts. Gendered impacts were underexplored within reviews. Reviewed literature underscored the value of engaging diverse knowledge systems; platforming localized, community-led adaptation to climate change and biodiversity loss, while addressing sociopolitical constraints to these efforts; and applying a broader conceptualization of health that aligns with Indigenous frameworks. Going forward, we must foreground equity- and rights-based considerations within integrated responses to climate and biodiversity crises.

**Funding:** Funding for this work came from a grant to WHO from the Government of Canada entitled "Strengthening local and national Primary Health Care and Health Systems for the recovery and resilience of countries in the context of COVID-19." The work was commissioned to the University of Waterloo through Agreement for Performance of Work 203050038_01. The funders had no role in study design, data collection and analysis, decision to publish, or preparation of the manuscript. This article represents solely the views of the authors and in no way should be interpreted to represent the views of, or endorsement by, WHO. WHO shall in no way be responsible for the accuracy, veracity, and completeness of the information provided through this article.

**Competing interests:** The authors have declared that no competing interests exist.

## Introduction

Enhancing the health and wellbeing of Indigenous Peoples amid the climate crisis and rapid biodiversity losses represents one of the most pressing and complex contemporary challenges globally [1–3]. Indeed, Indigenous Peoples, for whom the health of the land, environments, and all species are inextricably linked [4–6], are among those most disproportionately and acutely affected by the impacts of these ecological changes [7,8]. Equity and rights-based considerations are thus foregrounded in critical analyses of the climate crisis, biodiversity loss, and Indigenous health and wellbeing [1,9,10].

As the impacts of the climate crisis on Indigenous health and wellbeing are increasingly documented by scholars and practitioners worldwide, global discourse and calls for action have tuned to this interface of climate change and Indigenous health, as reflected in the 2022 report of the Intergovernmental Panel on Climate Change [11] and other global landmark reports [12]. In parallel, the links between biodiversity loss and Indigenous health are highlighted by global mechanisms such as the Intergovernmental Science-Policy Platform on Biodiversity and Ecosystem Services [13] and the Convention on Biological Diversity (i.e., see Kunming-Montréal Global Biodiversity Framework) [14]. There exists recognition of interconnectivity between these spheres of climate change, biodiversity loss, and Indigenous health and wellbeing [15]. Yet, a gap exists in research that cohesively and critically examines the interrelated impacts of climate change and biodiversity loss–connections of particular significance to Indigenous Peoples' livelihoods and wellbeing–and which aims to do so through a strengths-based lens [16]. Through a more comprehensive synthesis of these global literatures, we might identify Indigenous-led responses–opportunities for research, policy, and praxis–that advance Indigenous health and wellbeing, alongside broader ecological and planetary health. In this space, Indigenous voices, knowledges, and rights must be centred and prioritized, to drive future action with equity at the forefront [8,9].

Given the broad conceptual scope, as well as the expansive literatures at the climate-health nexus, a systematic umbrella review ('review of reviews') was chosen as the appropriate knowledge synthesis methodology [17,18]. The overarching research question guiding the review was as follows: What are the pathways through which climate change and biodiversity loss intersect with Indigenous health and wellbeing, as reported in the global secondary literature? To guide and deepen this inquiry on pathways of impact, we mapped climate change and biodiversity loss impacts against an adapted version of Neufeld et al. (2022)'s three-level framework. In this adapted framework, proximal impacts refer to direct impacts on physical health; intermediate impacts are those related to broader ecosystem changes; and distal impacts relate to culture-wide changes [11]. Indigenous health was defined and operationalized in alignment with Indigenous conceptualizations of health as inclusive of mental, emotional, spiritual, and physical wellbeing, and intrinsically tied to the land, land-based livelihoods, language, culture, and relationships [4,5,7].

Overall, this review aimed to characterize the extent, range, and nature of secondary literature on climate change, biodiversity loss, and Indigenous health and wellbeing globally. Based on the synthesized published and unpublished global literature, an objective of this review was to examine the proximal, intermediate, distal, and gendered impacts of climate change and biodiversity loss on Indigenous health and wellbeing. The final objective was to identify responses to climate change and biodiversity loss that also advance Indigenous health and wellbeing and reinforce the United Nations Declaration on the Rights of Indigenous Peoples (UNDRIP).

## Methods

### Rationale and conceptual foundation for the review

Recognizing the scale and significance of climate-health and biodiversity loss impacts on Indigenous Peoples' health and wellbeing, the Health Equity area of the World Health Organization (WHO) Headquarters' Gender, Equity and Human Rights Department, in collaboration with the WHO Headquarters' team for Biodiversity, Climate Change and Health, identified the need for a comprehensive literature review to support forthcoming work on climate change, biodiversity, and health. The review was commissioned to the Waterloo Climate Institute and conducted by a multidisciplinary team of Indigenous and non-Indigenous scholars at the Universities of Waterloo and Guelph, Canada.

The purpose of the review was to contribute to the health section of the forthcoming 2024 United Nations Department of Economic and Social Affairs State of the World's Indigenous Peoples report focused on climate change; to inform ongoing work for WHO with Member States and partners on climate change and health; and to contribute evidence towards implementing World Health Assembly resolution 76.16 on the "Health of Indigenous Peoples". This resolution calls for the creation of a Global Plan of Action on Indigenous Health, support to Member States on Indigenous Health, and integration of a focus on Indigenous health into the WHO's 14th General Programme of Work [19]. Moreover, this review aimed to inform the "Expert Working Group on Biodiversity, Climate, One Health, and Nature-Based Solutions" and support WHO regional-level capacity-building workshops on biodiversity and health.

### Study context

A systematic umbrella review was conducted in accordance with the Preferred Reporting Items for Systematic Reviews and Meta-Analyses (PRISMA) [20] (S1 File). Additional study details are reported in our protocol [21], as well as reported via 24 April 2023 registration with the International Prospective Register of Systematic Reviews (PROSPERO) (No. CRD42023417060).

All stages of this review were guided by an advisory committee of Indigenous experts, scholars, and civil society organization representatives, convened by WHO, as well as WHO staff responsible for commissioning the work. WHO organized two virtual engagement sessions with this group who were involved in the co-development of a research plan and methodological approach (February 2023 meeting) and provided critical feedback on preliminary results (July 2023 meeting).

### Search strategy and article selection

On 9 February 2023, the following five databases were searched for published academic secondary literature: Web of Science, Scopus, PubMed, CINAHL (via EBSCOHost), and the Campbell Collaboration. The sensitivity of the search approach was enhanced by a manual search of the following journals: *The Journal of Climate Change and Health; Environmental Health Perspectives; The Lancet Planetary Health; International Journal of Circumpolar Health; Anthrosource; AlterNative*; and the *International Journal of Indigenous Health*. The detailed search strategy and eligibility criteria are reported in the protocol [21].

In addition, the research team searched for relevant global reports, working papers, and policy briefs available in English in the United Nations databases; using an NGO/IGO search tool [22]; and through targeted Google searches (i.e., using combinations of key terms such as "Indigenous Peoples", health, and climate) from 28 March 2023 to 3 April 2023. A template was developed based on the methods outlined by Godin et al. (2015) [23]. This search

of the unpublished grey literature was targeted geographically to areas less represented in the published academic literature retrieved (i.e., Australia, New Zealand, Oceania, Latin America, Caribbean, Africa, South-East Asia, Middle East), as well as thematically, based on gaps identified through preliminary data extraction from the published peer reviewed literature.

Two independent reviewers (TC, LJB, or HN) conducted level one (abstract) and level two (full-text) eligibility screening of published academic literature using *Covidence*, a web-based systematic review platform. Included articles were: secondary sources; focused on Indigenous Peoples' health or wellbeing; and examined health/wellbeing in relation to climate change, biodiversity, and/or environment. Records meeting all inclusion criteria proceeded to data extraction.

### Data extraction and analysis

Two independent reviewers (TC, LJB, LP, or HN) extracted data according to pre-determined and piloted domains [21]. Each published article was appraised for methodological quality, using an adapted, composite tool based on the Joanna Briggs Institute and Critical Appraisal Skills Program checklists, as well as relevance and usefulness to Indigenous Peoples' priorities and processes. All discrepancies in judgment were discussed and resolved by consensus.

Basic descriptive statistics (proportions) were calculated to evaluate the extent, range, and nature of included literature. Qualitative data from quality appraisal and extraction processes (S1 and S2 Tables) were analyzed thematically, using a constant comparative method [24]. Initial observations of the data were discussed collaboratively as a research team. Building from these insights, two team members (LJB, LP) conducted detailed line-by-line coding of the extraction and quality appraisal frameworks, using *NVivo Release 1.5* software for organization and retrieval of codes and coded excerpts. Case attributes and classifications corresponding to data extraction and quality appraisal domains were applied to articles within *NVivo* to enable querying of co-occurrence of codes across reviews and to identify potential trends in the data (e.g., co-occurrence of a given 'geographic location' attribute with a high level of Indigenous Peoples' involvement in the research; or a 'study discipline' attribute with the type of climate responses recommended by articles). Emerging findings were shared with the expert advisory group, whose interpretations of the data, insights, and observations were integrated into the finalized results. Collaboration among the research team and with the advisory group enhanced the validity and rigour of the analyses [25].

### Results

### Describing the extent, range, and nature of secondary literature on climate change and Indigenous Peoples' health and wellbeing globally

Of 3156 published records retrieved from database searches, 38 met the eligibility criteria for inclusion (Fig 1). Records were published from 2010–2023, and reviewed literature back to 1956 (Tables 1 and 2). Based on the information available, the corpus of review articles synthesized searched a cumulative 37 databases, reviewed a total of 3575 records, and synthesized qualitative (71%), quantitative (68%), mixed methods (61%) and other primary research methodologies (e.g., case studies; 50%). Records reviewed literature across international, interdisciplinary databases including Indigenous databases, health, agriculture, food science, technology, biology, ecology, and zoology databases. Published literature covered a wide range of topics, largely within and cross-cutting the disciplines of public health, global health, environmental science and governance, and geography. The majority of records focused on the intersection of climate change, adaptation, and environmental/biodiversity loss with

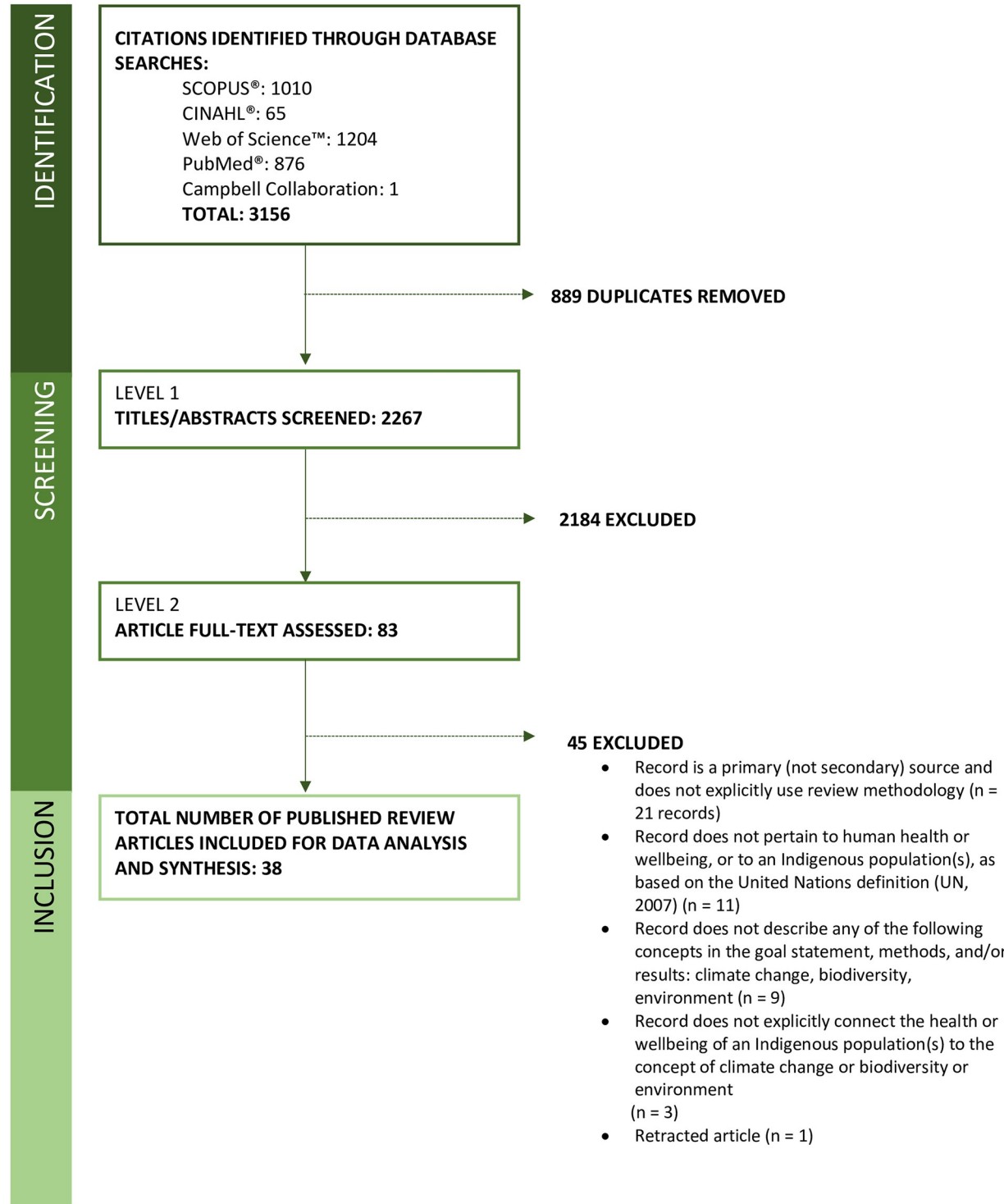

**Fig 1. PRISMA flow diagram, indicating the number of records retrieved from database searches; screened; and synthesized for inclusion in the review.**

**Table 1. List of included articles retrieved from a systematic search of the published academic secondary literature.** Articles are organized chronologically by year of publication. The author(s), title, study region(s), and review methodology are also indicated for each retrieved article.

| Year | Author(s) | Title of Article | Region(s) | Review Methodology |
|------|-----------|------------------|-----------|--------------------|
| 2010 | Ford, J.D., Berrang-Ford, L., King, M. and Furgal, C. | Vulnerability of Aboriginal health systems in Canada to climate change | North America (not Circumpolar) | Literature review |
| 2011 | Pearce, T., Ford, J.D., Duerden, F., Smit, B., Andrachuk, M., Berrang-Ford, L. and Smith, T. | Advancing adaptation planning for climate change in the Inuvialuit Settlement Region [ISR]: A review and critique | North America (Circumpolar) | Literature review |
| 2012 | Ford, J.D. | Indigenous health and climate change | Global | Systematic |
| 2012 | Ford, J.D., Bolton, K., Shirley, J., Pearce, T., Tremblay, M., Westlake, M. | Research on the human dimensions of climate change in Nunavut, Nunavik, and Nunatsiavut: A literature review and gap analysis | North America (Circumpolar) | Systematic |
| 2012 | Ford, J.D., Bolton, K., Shirley, J., Pearce, T., Tremblay, M., Westlake, M. | Mapping human dimensions of climate change research in the Canadian Arctic | North America (Circumpolar) | Systematic |
| 2014 | King, U. and Furgal, C. | Is hunting still healthy? Understanding the interrelationships between Indigenous participation in land-based practices and human-environmental health | Global | Scoping |
| 2015 | Loring, P.A. and Gerlach, S.C. | Searching for progress on food security in the North American North: A research synthesis and meta-analysis of the peer-reviewed literature | North America (Circumpolar) | Meta-analysis |
| 2016 | Jasmine, B., Singh Y., Onial, M. and Mathur, V.B. | Traditional knowledge systems in India for biodiversity conservation | Asia | Literature review |
| 2018 | Jaakkola, J.J.K., Juntunen, S. and Näkkäläjärvi, K. | The holistic effects of climate change on the culture, well-being and health of the Saami, the only Indigenous People in the European Union | Circumpolar (beyond North America) | Systematic |
| 2019 | Lam, S., Dodd, W., Skinner, K., Papadopoulos, A., Zivot, C., Ford, J., Garcia, P.J. and Harper, S.L. | Community-based monitoring of Indigenous food security in a changing climate: Global trends and future directions | Global | Systematic |
| 2019 | Kipp, A., Cunsolo, A., Gillis, D., Sawatzky, A. and Harper, S.L. | The need for community-led, integrated and innovative monitoring programmes when responding to the health impacts of climate change | North America (Circumpolar) | Scoping |
| 2019 | Akearok, G.H., Holzman, S., Kunnuk, J., Kuppaq, N., Martos, Z., Healey, C., Mak-Kik, R., Mearns, C., Mike-Qaunaq, A. and Tabish, T. | Identifying and achieving consensus on health-related indicators of climate change in Nunavut | North America (Circumpolar) | Scoping |
| 2019 | Dannenberg, A.L., Frumkin, H., Hess, J.J. and Ebi, K.L. | Managed retreat as a strategy for climate change adaptation in small communities: Public health implications | Global | Other (no explicit typology) |
| 2019 | Markkula, I., Turunen, M. and Rasmus, S. | A review of climate change impacts on the ecosystem services in the Saami homeland in Finland | Circumpolar (beyond North America) | Literature review |
| 2020 | Van Bavel B., Ford, L.B., Harper, S.L., Ford, J., Elsey, H., Lwasa, S. and King, R. | Contributions of scale: What we stand to gain from Indigenous and local inclusion in climate and health monitoring and surveillance systems | Global | Systematic |
| 2020 | Bryson, J.M., Bishop-Williams, K.E., Berrang-Ford, L., Nunez, E.C., Lwasa, S., Namanya, D.B. and Harper, S.L. | Neglected tropical diseases in the context of climate change in East Africa: A systematic scoping review | Africa | Scoping |
| 2020 | Jones, R., Macmillan, R. and Reid, P. | Climate change mitigation policies and co-impacts on Indigenous health: A scoping review | Global | Scoping |
| 2020 | Ingemann, C., Hansen, N.F., Hansen, N.L., Jensen, K., Larsen, C.V.L. and Chatwood, S. | Patient experience studies in the Circumpolar region: A scoping review | Circumpolar (beyond North America) | Scoping |
| 2020 | Middleton, J., Cunsolo, A., Jones-Bitton, A., Wright, C.J. and Harper, S.L. | Indigenous mental health in a changing climate: A systematic scoping review of the global literature | Global | Scoping |
| 2021 | Hillier, S.A., Taleb, A., Chaccour, E. and Aenishaenslin, C. | Examining the concept of One Health for Indigenous communities: A systematic review | Global | Systematic |
| 2021 | Schlingmann, A., Graham, S., Benyei, P., Corbera, E., Martinez Sanesteban, I., Marelle, A., Solemany-Fard, R. and Reyes-Garcia, V. | Global patterns of adaptation to climate change by Indigenous Peoples and local communities: A systematic review | Global | Systematic |

(*Continued*)

**Table 1.** (Continued)

| Year | Author(s) | Title of Article | Region(s) | Review Methodology |
|------|-----------|------------------|-----------|--------------------|
| 2021 | McNamara, K.E., Westoby, R. and Chandra, A. | Exploring climate-driven non-economic loss and damage in the Pacific Islands | Oceania | Systematic |
| 2021 | Vogliano, C., Murray, L., Coad, J., Wham, C., Maelaua, J., Kafa, R. and Burlingame, B. | Progress towards SDG 2: Zero hunger in Melanesia–A state of data scoping review | Oceania | Scoping |
| 2021 | Little, M., Hagar, H., Zivot, C., Dodd, W., Skinner, K., Kenny, T.A., Caughey, A., Gaupholm, J. and Lemire, M. | Drivers and health implications of the dietary transition among Inuit in the Canadian Arctic: A scoping review | North America (Circumpolar) | Scoping |
| 2021 | Kiddle, G.L., Zari, M.P., Blaschke, P., Chanse, V. and Kiddle, R. | An Oceania urban design agenda linking ecosystems services, nature-based solutions, traditional ecological knowledge and wellbeing | Oceania | Critical review |
| 2021 | Eerkes-Medrano, L. and Huntington, H.P. | Untold stories: Indigenous knowledge beyond the changing Arctic cryosphere | Circumpolar (beyond North America) | Other (no explicit typology) |
| 2022 | Reis, J., Zaitseva, N.V. and Spencer, P. | Pressing issues of environmental health and medical challenges in Arctic and Sub-Arctic regions | Circumpolar (beyond North America) | Systematic |
| 2022 | Sahu, M., Chattopadhyay, B, Das, R. and Chaturvedi, S. | Measuring impact of climate change on Indigenous health in the background of multiple disadvantages: A scoping review for equitable public health policy formulation | Global | Scoping |
| 2022 | Shafiee, M., Keshavarz, P., Lane, G., Pahwa, P., Szafron, M., Jennings, D. and Vatanparast, H. | Food security status of Indigenous Peoples in Canada according to the 4 pillars of food security: A scoping review | North America (not Circumpolar) | Scoping |
| 2022 | Borish, D., Cunsolo, A., Snook, J., Dewey, C., Mauro, I. and Harper, S.L. | Relationships between Rangifer and Indigenous well-being in the North American Arctic and Subarctic: A review based on the academic published literature | North America (Circumpolar) | Scoping |
| 2022 | Lebel, L., Paquin, V., Kenny, T.A., Fletcher, C., Nadeau, L., Chachamovich, E. and Lemire, M. | Climate change and Indigenous mental health in the Circumpolar North: A systematic review to inform clinical practice | Circumpolar (beyond North America) | Systematic |
| 2022 | Leal Filho, W. Totin, E., Franke, J.A., Andrew, S.M., Abubakar, I.R., Azadi, H., Nunn, P.D., Ouweneel, B., Williams, P.A. and Simpson, N.P. | Understanding responses to climate-related water scarcity in Africa | Africa | Literature review |
| 2022 | Davis, K., Ford, J.D., Quinn, C.H., Mosurska, A., Flynn, M., IHACC Research Team and Harper, S.L. | Shifting safeties and mobilities on the land in Arctic North America: A systematic approach to identifying the root causes of disaster | North America (Circumpolar) | Other (no explicit typology) |
| 2022 | Cottrell, C. | Avoiding a new era in biopiracy: Including Indigenous and local knowledge in nature-based solutions to climate change | Global | Other (no explicit typology) |
| 2022 | Gupta, H., Nishi, M. and Gasparatos, A. | Community-based responses for tackling environmental and socio-economic change and impacts in mountain social-ecological systems | Global | Other (no explicit typology) |
| 2022 | Hagen, I., Huggel, C., Ramajo, L., Chacón, N., Ometto, J.P., Postigo, J.C. and Castellanos, E.J. | Climate change-related risks and adaptation potential in Central and South America during the 21st century | Latin America | Other (no explicit typology) |
| 2022 | Charnley, G.E.C., Kelman, I. and Murray, K.A. | Drought-related cholera outbreaks in Africa and the implications for climate change: A narrative review | Africa | Narrative review |
| 2023 | Zimmermann, S., Dermody, B.J., Theunissen, B., Wassen, M.J., Divine, L.M., Padula, V.M., von Wehrden, H. and Dorresteijn, I. | A leverage points perspective on Arctic Indigenous food systems research: A systematic review | Circumpolar (beyond North America) | Systematic |

Indigenous health and wellbeing. Two records were conceptually-focused (i.e., discussing the positioning of Indigenous Knowledges within broader movements in the scholarship, such as nature-based solutions or OneHealth) [26,27] (Table 2).

**Published secondary literature at the nexus of climate change and Indigenous Peoples' health and wellbeing is geographically clustered in the Circumpolar North.** Overall, 16 published review articles (42%) focused on the Circumpolar North [28–43], nine of which

**Table 2. Description of published review articles included in the umbrella review.**

| Type of data | Number of records (Proportion) |
|---|---|
| **Year of publication** | |
| 2010–2014 | 6 (16%) |
| 2015–2019 | 8 (21%) |
| 2020–2023 | 24 (63%) |
| **Review methodology** | |
| Systematic | 12 (32%) |
| Scoping | 12 (32%) |
| Meta-analysis | 1 (3%) |
| Critical | 1 (3%) |
| Literature review | 6 (16%) |
| Narrative | 1 (3%) |
| Other [i.e., not explicit] | 5 (13%) |
| **Topics[a]** | |
| Climate change impacts on Indigenous health and wellbeing | 17 (45%) |
| Environmental and biodiversity loss | 8 (21%) |
| Climate change adaptation to impacts | 6 (16%) |
| Food systems and nutrition | 5 (13%) |
| Monitoring and evaluation | 4 (11%) |
| Health systems | 3 (8%) |
| Climate-related risks | 3 (8%) |
| Mental health and wellbeing | 2 (5%) |
| Socio-ecological system impacts | 2 (5%) |
| Conceptual | 2 (5%) |
| **Geographic distribution** | |
| Global | 12 (32%) |
| Africa | 3 (8%) |
| Asia | 1 (3%) |
| North America [not Circumpolar] | 2 (5%) |
| North America [Circumpolar-focused] | 9 (24%) |
| Circumpolar [not limited to North America] | 7 (18%) |
| Latin America | 1 (3%) |
| Oceania | 3 (8%) |

[a] Categories of topics are not mutually exclusive [i.e., a given record may be included in more than one category].

focused specifically on Circumpolar North America, and particularly Inuit Nunangat. The range of geographic locations included Africa [44–46]; Asia [47]; Oceania [48–50]; North America (not Circumpolar) [51,52]; and Latin America [53] (Table 2).

Reviews focused on the wellbeing of a wide geography of Indigenous Nations, groups, or organizations in their respective research studies. Only one record was explicitly focused on the experiences of urban Indigenous Peoples [51], with the majority of records focused on rural and/or remote locales. In total, five reviews were included as per a weighted criteria [see protocol, [21]] and covered Africa [44–46]; Latin America [53]; and Indigenous groups who broadly identified as being from mountainous regions [54]. In addition, 12 reviews focused on Indigenous Peoples' health and wellbeing globally (Tables 1 and 2).

The grey literature search retrieved 37 relevant publications consisting of policy briefs, discussion papers, organization reports, and media content (e.g., presentations, website

resources) from across Africa, Latin America and the Caribbean, Australia, New Zealand, Oceania, and Asia. Ten publications (27% of grey literature) were global reports retrieved from United Nations (UN) databases (S3 Table).

## Examining the proximal, intermediate, and distal impacts of climate change and biodiversity loss to the health and wellbeing of Indigenous Peoples

Land and place are central to Indigenous Peoples' lives and livelihoods, and thus the health and wellbeing impacts being experienced from climate and environmental change. For Indigenous Peoples, as Davis et al. (2022) highlight, *"the land is the heart of cultural and community life"* [38:2]. Just as the reviewed literature points to the centrality of land and place to Indigenous Peoples, so too does it indicate that *"impacts, adaptation, and vulnerability are highly place- and culture-specific"* [55:1263] and that the *"health of people and place demand an integrated engagement"* [56:5772].

Indeed, threaded through the biodiversity, proximal-intermediate-distal, and gendered thematic sections that follow is recognition that the health-related impacts of the climate crisis and biodiversity loss are embedded in, and inextricably tied to, Indigenous Peoples' connections to place. These impacts relate to both natural and built places (e.g., healthcare facilities; physical infrastructure) [31,51,57]. As conveyed in the sections to follow, across reviews, opportunities for response and pathways for advancing Indigenous Peoples' health and wellbeing were linked to land and place.

**Biodiversity impacts: Globally, ecosystems are changing in place-specific, localized ways.** The reviewed literature invariably discussed ecosystem changes and specific impacts on biodiversity as a result of climate change. Ecosystem changes were often described as resulting from human activities based in capitalism and colonialist systems that drive broader climatic and environmental change. Two reviews (5.3%) integrated climate change and biodiversity impacts within their overall framing or focus [36,54]. When discussed, changes to biodiversity were characterized broadly as ecosystem changes, encompassing both changes to physical environments and the plant and animal species therein.

Reported changes to physical environments were extensive and severe, characterized as *"[the risk of] large scale ecological transformation"* [53:12] and *"dramatically declining"* [49:3]. Aquatic [e.g., 48,49], atmospheric [e.g., 33], and terrestrial [e.g., 41,53] changes were described, as well as, most notably, cryospheric changes across Arctic and Subarctic regions (e.g., ice depth, extent, and timing of break-up/freeze-up) [e.g., 32,35,37,43,58,59] (Table 3).

These changing environments were linked to changes to wildlife and plant ecology, across the reviewed literature, including changes to species important for Indigenous food systems, medicines, and–more broadly–livelihoods. The overall health, abundance, distribution, migratory patterns, and predator-prey balance of key animal species relied upon for Indigenous Peoples' livelihoods were reportedly altered with climatic/environmental change [40,41,51], including that of marine mammals (e.g., ringed and hooded seals, narwhal, polar bears) [28,30,33,39], *Rangifer* species (e.g., reindeer, caribou) [29,36,60], birds (e.g., wild ptarmigan) [29,42,59], and fish (e.g., char) [30,34,35,42,52], some of which were referred to as 'keystone species' for ecosystem services and functioning, as well as Indigenous Peoples' livelihoods and wellbeing (e.g., bees in Melanesia [49], and *Rangifer* in North America [36]). Changes to the health, productivity, and *"agrodiversity"* of both cultivated and wild plant species (e.g., berries, medicinal plants) were also noted [35,43,48,52,54:1131], alongside an increase in invasive plant species that alter agriculturally significant crops and trees [49].

**Table 3. Type of water-, air-, and land-related effects of climate change, as discussed in the reviewed published literature, and associated references.**

| Type of factor | Specific effects discussed | Number of records (Proportion) | References |
|---|---|---|---|
| **Water-related** | Extreme, unpredictable weather events | 16 (42%) | [28,31,32,35,38,39,42,46,48–51,53,56–58] |
| | Changing ice conditions | 13 (34%) | [28,30,32,34,35,37–39,41,52,55,56,60] |
| | Precipitation | 8 (21%) | [42,45,49–53,58] |
| | Drought | 6 (16%) | [38,44,46,53,58,60] |
| | Flooding | 5 (13%) | [38,41,53,57,58] |
| | Lower inland water levels | 3 (8%) | [41,52,56] |
| | Sea level rise | 6 (16%) | [49,50,53,55,56,58] |
| | Warming waterways | 2 (5%) | [49,50] |
| | Decreased water quality | 6 (16%) | [32,44,46,50,53,57] |
| | Glacier retreat | 1 (3%) | [53] |
| **Air-related** | Changing temperature | 17 (45%) | [26,28,29,33,34,40–43,45,46,48,51,53,55,58,59] |
| | Humidity | 3 (8%) | [33,53,58] |
| | Air quality | 5 (13%) | [32,33,43,48,58] |
| | Altered seasons | 4 (11%) | [35,38,53,60] |
| | Atmospheric changes | 2 (5%) | [33,56] |
| **Land-related** | Changes in snow composition | 7 (18%) | [28,41–43,52,53,56] |
| | Permafrost thaw | 4 (11%) | [33,39,53,56] |
| | Vegetation changes | 3 (8%) | [52,58,60] |
| | Landscape hazards: erosion, landslides | 9 (24%) | [28,33,35,39,42,53,55,56,58] |
| | Changing contaminants exposure | 2 (5%) | [40,52] |
| | Changing exposure due to ice melts | 1 (3%) | [28] |
| | Algae blooms | 1 (3%) | [32] |

Overall, across geographic regions, the reviewed literature reported vast ecological changes that have impacted livelihoods [61,62], altered community dynamics [61,63,64], and disrupted knowledge-sharing practices [63,65,66]. As noted by Leal Filho and colleagues, degradation of land and biodiversity loss has had *"cascading effects"* on local people relying on ecosystem services [46:7].

**Proximal impacts: Ecosystem changes are challenging Indigenous Peoples' health outcomes.** A total of 33 reviews (86.8%) identified and described the proximal impacts of observed ecosystem changes on human health, often tied to higher temperatures and variable precipitation. Records identified an overall increase in disease susceptibility [43,51,58] and mortality [32,46], alongside an array of acute and chronic physiological impacts, such as increased heat stress and prevalence of cardiovascular diseases [48,51], respiratory illnesses like asthma and airborne diseases [32,33,43,48,51,58], and increased UVB exposure [56]. Mental health challenges were also reported, including emotional responses of worry, sadness, anger, and emotional distress [32,35,58,60].

An increase in infectious diseases was also identified in 19 articles (50%), including vector-, food-, and waterborne diseases, which were observed across geographies [26,28,31–33,39,42,44–46,48,50,51,53,56–59]. Foodborne diseases (e.g., E. coli, botulism, salmonella, trichinella, brucellosis) were found to be particularly significant concerns among Indigenous communities in Circumpolar North America, where temperature changes compromise traditional food storage methods [32,51,55,59]. Moreover, several reviews underscored challenges to healthcare access and high quality infrastructure in rural areas, exacerbating the impacts of infectious disease prevalence due to climate change [44,45,53,55].

Beyond threats to food safety from foodborne illnesses, nutrition-related impacts were reported in 15 reviews (39.4%). Micronutrient deficiencies were reported across geographies [44,48,49,57,58]. Climate-driven food insecurity negatively impacts diet quality, particularly in relation to traditional foods access [30,52,55]. This shift away from traditional foods, known for their nutrient density [31], has invariably led to an increased reliance on market foods, resulting in increased prevalence of metabolic conditions and nutrition-related diseases [30,34]. Further, changing exposures and sensitivities to contaminants in food sources were also reported through bioaccumulation in the food chain [28,32–34,40,41,52,56].

Finally, 17 articles (44.7%) identified that ecosystem changes globally are creating threats to human safety. Most notably, natural calamities or environmental hazards such as flooding were known to increase accidental injury or death [32,35,38,48,50,58]. Subsequent erosion, particularly in coastal regions of Latin America and the Circumpolar North, was also tied to these health risks by causing infrastructure instability [42,53]. While relocation was discussed as an adaptation strategy to these risks by two reviews [53,57], Dannenberg et al. (2019) noted that injury can occur before, during, and after these managed retreat actions. In the Circumpolar North, literature discussed the danger of unstable ice conditions or unpredictable weather patterns for people travelling on the land to engage in activities such as hunting, harvesting, or herding [28,30,37–39,43,56].

**Intermediate impacts: Ecosystem changes are changing human environments.** Across the secondary literature (n = 32; 84.2%), ecosystem changes were linked to intermediate impacts on human environments and systems of livelihoods. These included challenges to food and water systems as well as local economic and built environments.

Climate change impacts on Indigenous food and water systems were discussed in 28 reviews (73.6%) and recognized as global phenomenon [67]. Many studies focused on harvesting-related vulnerabilities [39] that constrain subsistence activities such as hunting, fishing, and foraging [32,35,39,43,52,55]. Across geographies, agroecosystem productivity constraints were identified, which significantly reduced land productivity, food and medicine diversity, and drinking water quality and availability [29,42,48,49,52,54,58,59]. Water scarcity was a particular concern in the Global South (Latin America, Africa, South Asia, Oceania) and seen as a key driver of food insecurity and disease [44,46,48,53,54]. Ecosystem changes affected the access and availability of traditional food sources and exacerbated existing food insecurity prevalence [30,39,52,55,57], changing traditional food cultures [28].

Economic environment challenges were also described by 11 reviews (28.9%), particularly as climate change and associated biodiversity loss disrupts economic activities that are part of the subsistence economy [28]. Indigenous food systems and their security hold significant economic benefits to Indigenous communities around the world [67]. Yet, as seen in the Circumpolar North, hunting activities require more economic resources to adapt to new challenges, exacerbating social inequalities [28,35,40,43,52,56]. Beyond food, a reduction in other natural resources, such as fuel wood and non-timber forest products, were observed to impact other parts of the world where these resources make important contributions to local economies [29,32,44,54,68]. These challenges also impacted agricultural livelihoods, as reported in multiple grey literature reports [61,69].

More broadly, 12 reviews (31.5%) reported how ecosystem changes are altering built environments. For instance, land changes such as erosion and permafrost melt deteriorate transportation routes [32,51,52], lead to loss of housing and secure shelter [48], and threaten cultural sites [28,51]. In turn, the integrity of community infrastructure and community viability are also threatened [32,33,37,49]. Notably, most reviews that included built environment impacts focused on health care access or system deficiencies caused or amplified by extreme weather events or infrastructure damage [31,32,44,48,55,57]. These changes have had

profound impacts on communities, necessitating climate-induced mobility, migration or relocation, as per the global literature [35,48,49,51,53,58].

**Distal impacts: Human environment changes are impacting Indigenous Peoples' relationships to place, culture, and each other.** Changes to human environments and livelihood systems were shown to have distal consequences on Indigenous Peoples' ways of life globally. Specifically, 27 review articles (71.1%) indirectly associated ecosystem changes with interpersonal and relational changes tied to spiritual and family life, oral history, and culture [55].

Sixteen reviews (42.1%) identified these relational challenges as concerning a person's sense of place, kinship, and identity and disrupting relationships with the land [e.g., 35,43,56]. Ecosystem changes that alter physical landscapes were reported to be especially significant to Indigenous contexts, because *"when physical landscapes change, stories, memories, or meanings may also change or fade away"* [29:1079]. Several articles reported reduced cultural, historical, or social, or physical ties to the land [28,29,35,36,38,48,57,60] with implications for community-level kinship [36]. For example, challenges to traditional food access and availability threatened the viability of food sharing networks and other practices dependent on community collaboration [30,34,39,52].

Traditional knowledge and cultural practices were also indirectly associated with ecosystem changes in 23 reviews (60.5%). Articles reported impacts on traditional knowledge in terms of loss, disruption, reduced relevance, and unreliability in its use and transmission [26,27,29,30,35,37,38,43,48,49,51,52,54,57,60]. In reviews focused on the Circumpolar North, there was notable concern for the transmission of this knowledge to younger generations given the observable erosion of land skills and institutional memory [28,32,37,39,58]. Ecosystem changes were reported to disrupt daily activities [57] and, in turn, contribute to 'culturecide' [48].

Changing human environments were also found to challenge the fabrics of Indigenous cultures by indirectly advancing cultural shifts or negotiations. Overall, five reviews (13.1%) reported changes in traditional practices or networks [28,35,40,43,49], such as hunters requesting cash payment for traditional foods [28,43]. Some articles drew attention to other contextual forces, such as economics and the effects of modernization (e.g., nutrition transition, wage-based economy), as compounding these impacts and forcing people to negotiate needs [28,40,49].

Many of these distal changes contributed to emotional and psychological health challenges, as identified in 15 reviews (39.4%). Some reviews characterized ecosystem changes as constraining Indigenous Peoples' abilities to engage with the land in ways that are necessary to sustain mental and emotional wellbeing [26,32,34–36,41,42,48,58,60]. Several reviews also discussed an increase in social pathologies (e.g., family violence, addiction, poverty, suicide) [51,56] and interpersonal stress, conflicts, and intrafamilial tension [35,46,57,58], sometimes over resource rights or a result of resource deficits [46,57].

Four review articles (10.5%) discussed the impact of climate change on Indigenous sovereignty and self-determination. These impacts included worries of loss of autonomy and Inuit sovereignty in the Circumpolar North [30,35], or the loss of nationality and state sovereignty in other global regions [48]. As Middleton et al. (2020) identified, ecosystem changes can be perceived to limit people's ability for self-determination, *"such that climate change was framed as a driver of 'environmental dispossession'"* [60:11]. In certain contexts, Indigenous populations were reported to be not only vulnerable to climate change, but also to policies of climate mitigation that put livelihoods at risk [70]. Grey literature reports highlighted circumstances of Indigenous Peoples' eviction from ancestral lands [61] as well as how livelihood changes affect Indigenous Peoples' legal, cultural, and spiritual obligations to care for ancestral lands and waters [14].

### The gendered impacts of climate change on Indigenous Peoples' health and wellbeing are rarely discussed in the published secondary and grey literature

Eighteen review articles (47.4%) briefly mentioned sex and/or gender in some way in relation to climate change impacts or biodiversity loss; however, sex and/or gender were not a focus of the research or results in these articles. Across this literature, gender dynamics were discussed within a male-female binary, with no reported inclusion of gender-diverse participants or broader acknowledgement of gender diversity within the text. No articles conducted a specific gender-based analysis, or stratified analysis by sex or gender variables.

The available literature introduced the presence of differing health-related risks and outcomes associated with climate change due to gendered household or community roles and responsibilities. For instance, globally, women often hold more caregiving responsibilities, which may increase proximity to climate-related communicable diseases [44,45]. Women and men may also experience differing mental, emotional, or psychosocial impacts resulting from altered roles due to climate/environmental change, such as loss of pride and self-worth among men whose hunting activities are limited [35]; or loss of social supports among women whose partners migrate for work [46]. Within certain cultural contexts, gender norms can also limit participation in activities like hunting, thus influencing access to resources like food [40,52,56].

Of the reviews that discussed sex and/or gender (n = 8; 44.4%), differing climate-related health outcomes were reported between men and women in the literature, though did not extensively explore possible explanations. For instance, women are reportedly at higher risk of metabolic conditions such as obesity and type 2 diabetes [42]; lower vitamin D levels and higher risk of iron deficiency [30]; lasting mental health impacts due to climate-related relocation [48]; and higher susceptibility to neglected tropical diseases [45]. Climate-induced food insecurity and food contamination were noted as impacting the health and micronutrient intakes of pregnant women [33,58]. The prevalence of negative mental health outcomes was also reportedly different among men and women, with one record reporting higher rates of suicide among men in a reviewed study [33], and another referencing increased symptoms of 'solastalgia' among women in response to observed/lived climatic changes [60]. Taken together, discussion of sex-related and/or gendered impacts in the reviewed literature mapped most closely to proximal (direct physical health) and distal (culture-wide) impacts of climate/environmental change.

When discussed in grey literature, reports highlighted climate-induced changes in knowledge-sharing practices and associated impacts on family structure [61] and other social challenges, linked in particular to the changing roles and responsibilities of women as holders of specific types of knowledge [71–73].

### Review articles suggest opportunities for responding to climate change in ways that also advance Indigenous health and wellbeing

An array of responses to these climate-health impacts were discussed across reviews: 'mechanical' responses such as enhanced monitoring/surveillance or warning systems [42,51,53]; regulatory responses [44,53,58,70], such as land-use policies and conservation legislation [54]; and, among all reviews, research responses. Across types of responses, however, there existed some cross-cutting themes.

**Engaging multiple, diverse knowledge systems supports adaptation to and deepens understandings of climate-health impacts.** Reviews characterized non-Westernized knowledge systems as: "Indigenous and Local Knowledge" [53,68]; "Traditional and Local

Knowledge" [54]; "Traditional Knowledge" [39,49]; "Traditional Ecological Knowledge" [50]; and Indigenous Knowledge (IK) [26,46,70]. Regardless of how it was termed, reviews broadly acknowledged the value of engaging multiple forms of knowledge, beyond Western systems, in response to climate change. However, reviews varied in how they positioned IK within this narrative and in relation to other knowledge systems (e.g., in *how* they proposed engagement with IK). Reviews posited a need to: '*integrate*', '*incorporate*', or '*include*' IK within Western knowledge systems [26,50,68] or resource management approaches [53]; to '*mobilize*' IK, in general [54]; to broadly *support Indigenous Knowledge-sharing* or '*intergenerational knowledge transmission*' [36,37]; to prioritize *"preservation or documentation of [Traditional and Local Knowledge] and practices"* [54:1133,55]; and to advance some form of '*knowledge co-production*' or '*bilateral information sharing*' (e.g., within partnerships between Indigenous Peoples and researchers, governments, or other organizations) [27,50,70]. Fewer articles explicitly positioned IK as integral or foundational to Western science understandings of climate-health impacts [58,70].

Nevertheless, engagement with diverse knowledge systems was recognized as deepening and broadening understandings of health-related climate change and biodiversity loss impacts [29,34,39] as well as *"improv[ing] quality of evidence about co-impacts"* [70:14], e.g., *"[through] collecting more responsive and representative data"* [59:17]. Many articles highlighted that ecosystems have always been changing to some extent, that *"adaptation to environmental change is a constant in their lives and they will continue to adapt"* [43:12] and that embedded in many IK systems is the ability to flex and adapt to these changes [34,38,46,55], with some reviews identifying IK itself as a *"determinant of adaptability/resilience"* to climate/environmental change [39:816]. In this light, reviews pointed to the need to both support community-led adaptation and also address broader sociopolitical constraints on community-determined responses.

**Continue to platform localized, Indigenous-led action for climate change adaptation.** Across the secondary literature, reviews characterized the needed response to climate change and biodiversity loss as being primarily at the locus of community (10 articles; 26.3%), rather than regional, national, or international responses (four; 10.5%), for effectiveness but also in alignment with normative values and principles. Reviews identified a need for *"community-driven"* responses [68:56]; strengthened *"community engagement"* [37,54:1138,70]; *"community centered discourse"* [27:167]; or 'community-based' or 'bottom-up' approaches [e.g., community-based monitoring] [41,67]. One review focused squarely on the voices and 'untold stories' of Indigenous Peoples with respect to climate change [43]. Others focused on *"community empowerment"* without explicit discussion of self-determination in community responses [54,55:1264]. Similarly, others applying a 'vulnerability framework' to analyze community adaptive capacity advocated for community-led responses; however, they did not go further to interrogate the processes, institutions, and structures that constrain these adaptation responses and underly existing vulnerabilities [28,37,39,51,55]. On the other hand, some reviews went further to underscore institutions that create structural violence and risk [38] and, more foundationally, to link localized health-related climate and biodiversity vulnerabilities to broader sociopolitical contexts and colonialism.

**Interrogate the structural, institutional, and processual constraints to Indigenous-led adaptation, explicitly naming colonialism where it exists.** Eighteen reviews (47.4%) explicitly situated their findings regarding the health-related impacts of climate change and biodiversity loss in the context of historic and ongoing colonialism and associated power imbalances (e.g., rated 'high' or 'medium' with respect to degree of attentiveness to colonial influences in quality appraisal) (S1 Table). Colonialism was acknowledged as an *"underlying root cause of vulnerability"* to climate/environmental change [55:1260], that *"there needs to be a shift from a*

*focus on 'vulnerable peoples' to the underlying processes and institutions that put people at risk"* [38:15]. Colonialism, in the form of land dispossession and degradation [49], *"top-down governance structures [and] inflexible policies regarding land use and resource management"* [40:387] and hegemony of Western knowledge systems and structures [31,42,55], was named as antecedent to and driver of climate-change vulnerabilities [43]. These anthropogenic factors reportedly challenged intergenerational knowledge sharing [30,56] and affected access to health services, as well as food and water safety and security [44,46,50], contributing to and compounding climate change-vulnerabilities. Three reviews (7.9%) named settler colonialism, specifically, in the framing of the review [26,42] and situating of results [38], whereas most of the reviews discussing colonialism used the term broadly (i.e., to possibly include other forms such as franchise or exploitation colonialism).

A total of 18 articles (47.4%) were rated 'low' in the quality appraisal process in terms of attentiveness to colonial influences on climate-health and biodiversity impacts. For some, this discussion was less relevant to the type/aim of the article [32,36], or colonization was briefly mentioned but not engaged with extensively [52]. Many reviews also identified broader health and social inequities–disparities in social determinants of health–that affect Indigenous Peoples' vulnerability to climate change and biodiversity loss impacts on health [44,49,51,53,55], including poverty, inaccessibility of health services, poor infrastructure and public service systems, and racial discrimination. These articles identified the need for responses (i.e., policy, research) to be targeted towards these broader health and wellbeing challenges, to buffer the impacts of climate and environmental change.

**Climate-health research is needed that is strengths-based, trans- and interdisciplinary, embedded in partnership with Indigenous Peoples, and that applies a broader health lens.**   Based on their respective literature syntheses, reviews identified the need for future climate-health research approaches that are trans- and interdisciplinary [32,34]; strengths-based [43,48]; and that apply a broader lens and conceptualization of health, reflective of Indigenous concepts of health and wellbeing [55,56]. A research response to climate-health and biodiversity impacts needs to engage the complexity of socio-ecological relationships [56]. Reviews proposed OneHealth [26], 'nature-based solutions' [27], and 'ecosystem-based adaptation' [50] as frameworks for inquiry with these embedded systems or relational lenses, or that apply an eco-centric view to the design and development of adaptation approaches, and further examined the interfacing or integration of IK into these Western-science frameworks or models. Cottrell proposed that *"nature-based thinking"* (e.g., focused on the intrinsic value of nature)–a more inclusive and expansive framework to climate response–could help bridge the 'divide' between the Western scientific community and Indigenous communities [27:167]. Moreover, practitioners of a 'nature-based solutions' approach, in particular, *"can support the sovereignty of Indigenous peoples by advocating for local management and control over project lands"* [27:167].

More fundamentally, reviews proposed that primary research studies: be embedded in partnership with Indigenous Peoples [32,36,40,51,54]; involve a bilateral sharing of information rather than the scientific community, *"providing unilateral advice"* [43:6]; and involve a *"sharing of power"* and privileging of IK [34,70:14], through diverse methodological approaches [60] and *"study design that fully embeds Indigenous values, realities, and priorities"* [70:14].

Among the reviewed secondary literature itself, however, only 6 records [15.8%] explicitly reported a high degree of involvement of Indigenous Peoples across the research process (S1 Table). These reviews were led by Indigenous scholars [70] or teams involving 'knowledge users' or community researchers from the study context [32,39]. Review methodologies were informed by Indigenous partners or a broader steering committee [36,60], or reviews reported that Indigenous Peoples had input on the final manuscript to ensure framing aligned with their priorities [30].

**Shift the narrative within climate-health adaptation from prioritizing community-based approaches to community-driven, rights-based approaches that emphasize Indigenous sovereignty and autonomy.** Reviews discussed the crucial need to advance priorities and processes related to Indigenous sovereignty, rights, and autonomy within the climate-health, planetary health space. Records identified that, across the primary literature, there existed recognition of a need to prioritize and respect Indigenous sovereignty and autonomy within climate-adaptation efforts, whether related to food systems [30,34,40,52], health systems [31,51], or monitoring and surveillance systems [35,41,67]. As Van Bavel et al. (2020) named, there exists "*an ethical practice gap in the recognition and actualization of Indigenous and local autonomy, intellectual property rights, and data sovereignty in integrated [monitoring and surveillance systems]*" as well as a need across the literature to move "*from inclusion to ownership*" [59:18].

This need extended to research and policy spheres, where reviews identified a gap in approaches that are rights-based [51,58,70], and noted that "*even the ability to define the problem on one's own terms represents in many places a move away from the status quo*" [40:388]. Moreover, a focus on rights within policy responses would reportedly look like "*local management and control over project lands*" [27:167], and "*greater Indigenous autonomy over mobility, time, education and land use*" [38:15], as well as "*a new ethos of coordination and cooperation among government levels*" to address institutional determinants of health [51:677]. Taken together, these findings are reflected in Loring & Gerlach's (2015) underscoring of a "*need to see beyond past concepts such as adaptation and resilience and look instead to rights-based reform*" [40:387]. Responses to climate change and biodiversity loss, then, need to interrogate the locus of power—question who is holding it—and move towards a model of equitable and shared decision-making [70], focused on supporting Indigenous Peoples' sovereignty, rights, and autonomy as outlined in the UNDRIP.

## Discussion

These findings highlight that the health-related impacts being experienced due to the climate crisis and biodiversity loss, and broader conceptualizations and experiences of wellbeing, are place-based—inextricably tied to land and place. This resonates with Indigenous Peoples' known, lived experiences and is also widely discussed within written scholarship [4,74]. Indigenous Peoples' ontologies and epistemologies are intricately connected with the land [8,75–77]. Moreover, Indigenous Peoples' pathways to wellness are through relationality, responsibility, and kinship with the land, viewed holistically as the "*combined living spirit of plants, animals, air, water, humans, histories, and events*" [3:7,76,77]. Supporting the advancement of Indigenous Peoples' rights to lands and territories not only contributes to the wellbeing of Indigenous Peoples but to addressing the broader, complex challenges posed by climate change and biodiversity loss [11,75,78].

The findings of this review further substantiate the warning that, within the context of ongoing colonization, we are at an ecological and relational tipping point [70]. This inquiry traced the impacts of colonialism—a distancing from land and place through institutional structures, practices, policies, and systems [10,79]—through to biodiversity loss and the proximal, intermediate, and distal health outcomes experienced by Indigenous Peoples globally. Indeed, empirical evidence across the scholarship indicates how structural determinants are filtering down to community and individual impacts on both human and non-human species.

Colonialism, including settler colonialism, is increasingly recognized as a determinant and driver of climate-health and biodiversity impacts [8,12,80,81]. This recognition is important, yet Indigenous scholars and advocates have indicated that it is unclear where planetary health

fits into the existing determinants of health language [8]. An ecocentric approach is needed, a critical repositioning of human engagement with the land and environments, in our understanding of determinants of health and wellbeing [75,80,82] and to better inform research, policy, and community-led responses. Given that only two reviews in this study integrated considerations of climate change and biodiversity loss (the majority focused on only one of these dimensions), opportunity exists for further integration of both climate change and biodiversity into Indigenous-led health research and adaptation responses more broadly. While responses identified in the secondary literature were primarily located at the community scale, broader global movements–through mechanisms such as the United Nations Permanent Forum on the Rights of Indigenous Peoples and the Expert Mechanism on the Rights of Indigenous Peoples–may serve to galvanize community-led efforts and may themselves be informed by bottom-up, community-focused work to address the climate crisis. Moreover, the United Nations Permanent Forum on the Rights of Indigenous Peoples has proposed a framework on Indigenous determinants of health to guide the United Nations and member states in strategy, policy setting, and actions [83].

Further gaps exist within the published secondary literature in studies that have a global scope or are geographically focused beyond the Circumpolar North. The high proportion of reviews (42%) focused on the Circumpolar North may have implications for the types of proximal, intermediate, and distal impacts identified, as well as the specific concerns around biodiversity loss described within our findings. Additionally, a limited body of literature examines the health and wellbeing of Indigenous Peoples living in cities or the gendered impacts of climate change and biodiversity loss on Indigenous Peoples' health and wellbeing. More research is needed that conducts in-depth gendered analyses of climate-health and biodiversity impacts, particularly studies that consider the perspectives and experiences of gender-diverse individuals along with self-identified women and men. Opportunity exists for research to take an intersectional approach and interrogate how additional factors such as income, employment, housing, age, and class intersect with and shape experiences of Indigenous health, gender, and climate change [84].

Finally, the reviewed literature outlined further opportunities for advancing Indigenous health and wellbeing in the context of climate change and biodiversity loss. Across geographies, this literature signalled a need to consider the historical, political, and geographic contexts of climate change, alongside the structures of power (i.e., Western, colonial, and capitalist worldviews) that constrain climate adaptation and responses and disrupt Indigenous Peoples' connection to lands and waters. This need to dissect and critically examine the broader context of structural and systemic determinants of climate-health and biodiversity impacts is increasingly the focus of global public health scholarship [85]. In this light, reviews underscored the need to continue to support, fund, and platform localized, community-led adaptation to climate change and biodiversity loss, while addressing broader sociopolitical constraints to Indigenous Peoples' community-determination and leadership. Place-specific, localized adaptation responses are needed, informed by local data and knowledge. Moreover, as echoed throughout the literature, engaging, centering, and acknowledging multiple, diverse, non-Westernized knowledge systems and values will support climate change adaptation and deepen understanding of climate-health and biodiversity loss impacts [86].

## Limitations

This review has several limitations. Importantly, the umbrella review methodology relies on already-synthesized insights across the published literature. This may have limited the depth and breadth of findings related to climate-health and biodiversity impacts, particularly if the

secondary literature is not as geographically or topically diverse as the primary. We also may not have captured the most current published literature through this review of reviews. We aimed to address these limitations through a robust search and integration of the unpublished and current 'grey' literature, focused particularly on geographic gaps in published reviews, as well as our application of a weighted criteria to include additional reviews from geographic regions or populations less represented in the published literature. In addition, the research team's positionality as Western-trained researchers analyzing literature largely embedded within Western knowledge systems, and published almost exclusively in English, is a notable limitation of this work, in that it limits the nuance and depth of understanding of climate-health and biodiversity loss impacts emergent from Indigenous frameworks and knowledge systems. The knowledge systems embedded in other cultures, disciplines, and sectors may provide rich evidence of climate-health and biodiversity loss impacts that differs from evidence generated through Western-oriented knowledge synthesis approaches. While relying on the perspectives of the Indigenous expert advisory group to shape this study, our positionalities and the epistemological domain our findings are firmly embedded within, remain a limitation.

## Conclusion

The findings of this review are threaded together by one central theme: that the impacts of climate change and biodiversity loss on health and wellbeing are rooted in and inseparable from Indigenous Peoples' connections to place. Because of this, place-specific environmental change and biodiversity loss are driving proximal, intermediate, and distal changes to Indigenous health and wellbeing. Through bringing together global literatures on both climate change and biodiversity loss impacts, with an eye to identifying Indigenous-led responses to these crises, this umbrella review presents opportunities for advancing Indigenous health and wellbeing alongside broader ecological and planetary health.

Taken together, these findings suggest that *context matters*. As reflected in reviews, the *temporal context* of colonialism, historic and ongoing, shapes understandings of the antecedents to and drivers of climate-health and biodiversity loss impacts for Indigenous Peoples. *Cultural and epistemological context matters*, as the inclusion and prioritization of diverse knowledge systems strengthens monitoring and adaptation to climate change and biodiversity loss. *Geographic context matters*, as sociopolitical processes at multiple scales constrain or support adaptation efforts. As per the reviewed literature, place-specific and localized responses are needed, informed by local data and knowledge, firmly embedded in partnership with, by, and for Indigenous Peoples, and in service to Indigenous rights, sovereignty, and autonomy as addressed by UNDRIP. In alignment with the reflection on UNDRIP of Redvers et al. (2023) [9], positioning this instrument more prominently across geographies, scales, sectors, and literatures may support the continued integration of climate and biodiversity-related responses, while foregrounding equity and Indigenous rights within the spheres of research, policy, and praxis.

## Supporting information

**S1 File. Completed PRISMA checklist.**
(DOCX)

**S1 Table. Excerpt of quality appraisal data chart.**
(DOCX)

**S2 Table. Summary of studies and extracted data pertaining to key findings, observations, and recommendations.**
(DOCX)

**S3 Table. Grey literature records retrieved and included in the review.**
(DOCX)

## Acknowledgments

We would like to express our sincere gratitude to the following Indigenous Experts (featured in alphabetical order by last name) who had a key role in co-shaping the review through their feedback during the process, including inputs to the design of the terms of reference for the work, inputs to the protocol for the review, written feedback on the manuscript, and/or sharing of related publications with the research team and WHO: Amina Amharech, Arthur Blume, Tonje Margrete Winsnes Johansen, Lena Maria Nilsson, Sara Olsvig, and Anne Simpson. Thank you also to the following Indigenous Experts who participated in one or both of the engagement meetings, providing valuable guidance: Mukaro Borrero, Phoolman Chaudhary, Samantha Chisholm Hatfield, Rhys Jones, Belkacem Lounes, Margaret Tunda Lepore, Casey Kickett, Aqqaluk Lynge, Hanieh Moghani, Sherry Pictou, Zaira Zambelli Taveira, and Tarcila Rivera Zea. We would like to thank the responsible WHO officers for commissioning and overseeing this work, Theadora Swift Koller and Cristina Romanelli, as well as Hortense Nesseler who contributed to reviewing articles and project management and WHO consultants Susana Gomez and Rachel Hammonds who contributed to the terms of reference and revised aspects of the final text, respectively. We would also like to thank Sarah Burch, Simon Glauser, and Chenel Lozon of the Waterloo Climate Institute for managing the contract, alongside providing logistical and administrative support. Thank you to India Bruton who supported the development of figures and tables for this manuscript. Thank you also to Jackie Stapleton at the University of Waterloo library for supporting the development of the search strategy.

## Author Contributions

**Conceptualization:** Laura Jane Brubacher, Sheri Longboat, Warren Dodd, Susan J. Elliott, Hannah Neufeld.

**Formal analysis:** Laura Jane Brubacher, Laura Peach, Tara Tai-Wen Chen.

**Funding acquisition:** Hannah Neufeld.

**Investigation:** Laura Jane Brubacher, Laura Peach, Tara Tai-Wen Chen, Kaitlyn Patterson, Hannah Neufeld.

**Methodology:** Laura Jane Brubacher, Sheri Longboat, Warren Dodd, Susan J. Elliott, Hannah Neufeld.

**Project administration:** Hannah Neufeld.

**Supervision:** Hannah Neufeld.

**Visualization:** Tara Tai-Wen Chen.

**Writing – original draft:** Laura Jane Brubacher, Laura Peach.

**Writing – review & editing:** Laura Jane Brubacher, Laura Peach, Tara Tai-Wen Chen, Sheri Longboat, Warren Dodd, Susan J. Elliott, Kaitlyn Patterson, Hannah Neufeld.

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
