## [Decision Letter · Decision Letter 0]

20 Dec 2023

PGPH-D-23-02151

Climate change, biodiversity loss, and Indigenous Peoples' health and wellbeing: A systematic umbrella review

Dear Dr. Brubacher,

Thank you for submitting your manuscript to PLOS Global Public Health. After careful consideration, we feel that it has merit but does not fully meet PLOS Global Public Health’s publication criteria as it currently stands. Therefore, we invite you to submit a revised version of the manuscript that addresses the points raised during the review process.

We look forward to receiving your revised manuscript.

Kind regards,

Prashanth Nuggehalli Srinivas, MBBS, MPH, PhD

Academic Editor

Journal Requirements:

2. We do not publish any copyright or trademark symbols that usually accompany proprietary names, eg  ©, ®, ™  (e.g. next to drug or reagent names). Please remove all instances of trademark/copyright symbols throughout the text, including ® & ™.

Additional Editor Comments (if provided):

Reviewers' comments:

Reviewer's Responses to Questions

**Comments to the Author**

1. Does this manuscript meet PLOS Global Public Health’s publication criteria? Is the manuscript technically sound, and do the data support the conclusions? The manuscript must describe methodologically and ethically rigorous research with conclusions that are appropriately drawn based on the data presented.

Reviewer #1: Yes

Reviewer #2: Yes

2. Has the statistical analysis been performed appropriately and rigorously?

Reviewer #1: Yes

Reviewer #2: Yes

3. Have the authors made all data underlying the findings in their manuscript fully available (please refer to the Data Availability Statement at the start of the manuscript PDF file)?

Reviewer #1: Yes

Reviewer #2: Yes

4. Is the manuscript presented in an intelligible fashion and written in standard English?

Reviewer #1: Yes

Reviewer #2: Yes

5. Review Comments to the Author

Reviewer #1: This is a brilliant review and synthesis of the available literature, with sound methodology and an appropriate and ethnical approach (i.e. Indigenous consultation and oversight). The breadth and detail of the findings are laudable, with a sound discussion that pulls together the threads and provides a clear path forward for future research. As a very minor suggestion - because settler colonial studies and elements of critical Indigenous studies have been at pains to distinguish settler colonialism from other forms (i.e. franchise or exploitation colonialism), and because there's a growing literature and focus on settler colonialism and health - it may be worth at least noting in summary statements/the discussion that you are including settler colonialism within the umbrella of colonialism. And of course if any of the papers specified or distinguished, this might be helpful to tease out as well. Thank you for this excellent, high-quality, and very useful addition to the literature.

Reviewer #2: All comments attached are for authors and editors. This submission needs little revision to be ready for publication. As far as I could tell, the availability of data will be met. This is a quality journal article submission.

6. PLOS authors have the option to publish the peer review history of their article (what does this mean?). If published, this will include your full peer review and any attached files.

**Do you want your identity to be public for this peer review?** For information about this choice, including consent withdrawal, please see our Privacy Policy.

Reviewer #1: **Yes: **Bram Wispelwey

Reviewer #2: No

---

## [Decision Letter · Decision Letter 1]

15 Feb 2024

Climate change, biodiversity loss, and Indigenous Peoples' health and wellbeing: A systematic umbrella review

PGPH-D-23-02151R1

Dear Dr. Brubacher,

We are pleased to inform you that your manuscript 'Climate change, biodiversity loss, and Indigenous Peoples' health and wellbeing: A systematic umbrella review' has been provisionally accepted for publication in PLOS Global Public Health.

Best regards,

Julia Robinson

Executive Editor

Reviewer Comments (if any, and for reference):

Reviewer's Responses to Questions

**Comments to the Author**

1. If the authors have adequately addressed your comments raised in a previous round of review and you feel that this manuscript is now acceptable for publication, you may indicate that here to bypass the “Comments to the Author” section, enter your conflict of interest statement in the “Confidential to Editor” section, and submit your "Accept" recommendation.

Reviewer #1: All comments have been addressed

Reviewer #2: All comments have been addressed

2. Does this manuscript meet PLOS Global Public Health’s publication criteria? Is the manuscript technically sound, and do the data support the conclusions? The manuscript must describe methodologically and ethically rigorous research with conclusions that are appropriately drawn based on the data presented.

Reviewer #1: Yes

Reviewer #2: Yes

3. Has the statistical analysis been performed appropriately and rigorously?

Reviewer #1: Yes

Reviewer #2: (No Response)

4. Have the authors made all data underlying the findings in their manuscript fully available (please refer to the Data Availability Statement at the start of the manuscript PDF file)?

Reviewer #1: Yes

Reviewer #2: (No Response)

5. Is the manuscript presented in an intelligible fashion and written in standard English?

Reviewer #1: Yes

Reviewer #2: (No Response)

6. Review Comments to the Author

Reviewer #1: This is an important publication - thank you for this work.

Reviewer #2: (No Response)

7. PLOS authors have the option to publish the peer review history of their article (what does this mean?). If published, this will include your full peer review and any attached files.

**Do you want your identity to be public for this peer review?** For information about this choice, including consent withdrawal, please see our Privacy Policy.

Reviewer #1: **Yes: **Bram Wispelwey

Reviewer #2: No
